# Hospitalizations for Acute Salicylate Intoxication in the United States

**DOI:** 10.3390/jcm9082638

**Published:** 2020-08-14

**Authors:** Charat Thongprayoon, Tananchai Petnak, Wisit Kaewput, Michael A. Mao, Karthik Kovvuru, Swetha R. Kanduri, Boonphiphop Boonpheng, Tarun Bathini, Saraschandra Vallabhajosyula, Aleksandra I. Pivovarova, Himmat S. Brar, Juan Medaura, Wisit Cheungpasitporn

**Affiliations:** 1Division of Nephrology and Hypertension, Department of Medicine, Mayo Clinic, Rochester, MN 55905, USA; 2Division of Pulmonary and Pulmonary Critical Care Medicine, Department of Medicine, Faculty of Medicine, Ramathibodi Hospital, Mahidol University, Bangkok 10400, Thailand; 3Division of Pulmonary and Critical Care Medicine, Department of Medicine, Mayo Clinic, Rochester, MN 55905, USA; 4Department of Military and Community Medicine, Phramongkutklao College of Medicine, Bangkok 10400, Thailand; 5Division of Nephrology and Hypertension, Department of Medicine, Mayo Clinic, Jacksonville, FL 32224, USA; mao.michael@mayo.edu; 6Department of Medicine, Ochsner Medical Center, New Orleans, LA 70121, USA; karthik.kovvuru@ochsner.org (K.K.); svetarani@gmail.com (S.R.K.); 7Department of Medicine, David Geffen School of Medicine, University of California, Los Angeles, CA 90095, USA; boonpipop.b@gmail.com; 8Department of Internal Medicine, University of Arizona, Tucson, AZ 85721, USA; tarunjacobb@gmail.com; 9Section of Interventional Cardiology, Division of Cardiovascular Medicine, Department of Medicine, Emory University School of Medicine, Atlanta, GA 30322, USA; saraschandra.vallabhajosyula@emory.edu; 10Division of Nephrology, Department of Internal Medicine, University of Mississippi Medical Center, Jackson, MS 39216, USA; apivovarova@umc.edu (A.I.P.); hbrar@umc.edu (H.S.B.); jmedaura@umc.edu (J.M.)

**Keywords:** salicylate, salicylate intoxication, epidemiology, hospitalization, outcomes

## Abstract

Background: The objective of this study was to describe inpatient prevalence, characteristics, outcomes, and resource use for acute salicylate intoxication hospitalizations in the United States. Methods: A total of 13,805 admissions with a primary diagnosis of salicylate intoxication from 2003 to 2014 in the National Inpatient Sample database were analyzed. Prognostic factors for in-hospital mortality were determined using multivariable logistic regression. Results: The overall inpatient prevalence of salicylate intoxication among hospitalized patients was 147.8 cases per 1,000,000 admissions in the United States. The average age was 34 ± 19 years. Of these, 35.0% were male and 65.4% used salicylate for suicidal attempts. Overall, 6% required renal replacement therapy. The most common complications of salicylate intoxication were electrolyte and acid-base disorders, including hypokalemia (25.4%), acidosis (19.1%), and alkalosis (11.1%). Kidney failure (9.3%) was the most common observed organ dysfunction. In-hospital mortality was 1.0%. Increased in-hospital mortality was associated with age ≥30, Asian/Pacific Islander race, diabetes mellitus, hyponatremia, ventricular arrhythmia, kidney failure, respiratory failure, and neurological failure, while decreased in-hospital mortality was associated with African American and Hispanic race. Conclusion: hospitalization for salicylate intoxication occurred in 148 per 1,000,000 admissions in the United States. Several factors were associated with in-hospital mortality.

## 1. Introduction

Salicylates have been used for more than 3000 years as an analgesic. Acetylsalicylate, also called aspirin, is currently used as an analgesic, antipyretic, and antiplatelet agent. In addition to its oral preparation, methyl salicylates are also found in topical analgesic products, such as ointments, herbal oils, and lotions [1]. Since most salicylate-containing products are over-the-counter medications, salicylate availability is widespread. As such, salicylate intoxication is one of the most common drug overdoses. The American Association of Poison Control Centers reports the annual burden of salicylate intoxication. In the United States in 2018, 27,644 patients were exposed to salicylates (both acetylsalicylate and methyl salicylate) with an associated 0.4% mortality rate. Intentional ingestion accounted for 52.8% of reported salicylate toxicity [2].

After ingestion, salicylates are rapidly absorbed in the stomach. Salicylates are mostly protein bound and therefore limited to intravascular space. Salicylates are metabolized via several different pathways in the liver, including glycination into salicyluric acid, which is both less toxic and more rapidly excreted by kidney. Only a small amount of salicylates is excreted unchanged in urine [1]. Salicylates directly or indirectly affect most organ systems by uncoupling oxidative phosphorylation, inhibiting Krebs cycle enzymes, and inhibiting amino acid synthesis [3]. Salicylate intoxication can be categorized as acute versus chronic salicylate intoxication. Acute intoxication is typically seen in younger patients who intentionally ingest high doses of salicylate-containing agents as a suicidal attempt [3]. Serum salicylate levels generally correlate with the severity of the disease. In acute intoxication, patients may present with tachypnea, nausea, vomiting, dizziness, fever, tinnitus, diaphoresis, altered mental status, seizure, coma, kidney failure, or hypotension [1]. In contrast, chronic salicylate intoxication is generally found in older patients who habitually use salicylates for chronic underlying illnesses. Since chronic salicylate intoxication results in slow tissue accumulation and saturation, including in the central nervous system, the manifestation of symptoms often does not correlate with serum salicylate levels [3]. Chronic intoxication symptoms may be absent or mild, and neurological symptoms may include tremor, restlessness, or behavioral alterations. Thus, patients with chronic salicylate intoxication may be misdiagnosed, particularly in the elderly or patients with multiple comorbidities and polypharmacy [1].

As salicylates remain widely available and serve as a cornerstone of many recommended medical treatments, knowledge of salicylate intoxication and its heterogeneous presentation, treatment, and sequelae is still required. However, large contemporary studies on salicylate intoxication have been limited. We conducted this cohort study to describe the inpatient prevalence, clinical characteristics, outcomes, and resource use for acute salicylate intoxication hospitalizations in the United States.

## 2. Methods

### 2.1. Study Population

The National Inpatient Sample (NIS) database is the largest all-payer database of hospitalized patients in the United States. The NIS database contains hospitalization data from a stratified sample of 20% of hospitals in the United States. The data consist of diagnosis and procedure codes. National estimates for hospitalization nationwide are estimated using a sample weight method. The institutional review board approval was waived because the de-identified data were publicly available. This study identified patients hospitalized primarily for salicylate intoxication using the International Classification of Diseases, Ninth Revision (ICD-9) diagnosis code of 965.1.

### 2.2. Variables and Outcomes of Interest

Clinical characteristics consisted of age, sex, race, period of hospitalization, alcohol drinking, suicidal attempt, analgesics overdose, psychotropic medication overdose, and certain comorbidities. Treatments recorded consisted of gastric lavage, non-invasive and invasive mechanical ventilation, and renal replacement therapy. Complications recorded consisted of hypoglycemia, electrolyte and acid-base disturbance, rhabdomyolysis, seizure, gastrointestinal bleeding, sepsis, and ventricular arrhythmia. Outcomes consisted of organ failure, which included failure of any of the following systems: renal, respiratory, circulatory, liver, neurological, and hematological systems. Outcomes also included in-hospital mortality. Resource use consisted of hospitalization cost and length of hospital stay. The ICD-9 codes for comorbidities, treatments, complications and outcomes are shown in Appendix A.

### 2.3. Statistical Analysis

Descriptive statistics were used, as appropriate, to summarize the clinical characteristics, treatments, outcomes, and resource use for hospitalization for salicylate intoxication. The United States’ annual inpatient prevalence of salicylate intoxication from 2003 to 2014 was determined. Prognostic factors for in-hospital mortality were determined using multivariable logistic regression analysis with backward stepwise selection. The statistical analysis was significant when two-tailed *p*-value < 0.05. The statistical analysis was performed using SPSS statistical software (version 22.0, IBM Corporation, Armonk, NY, USA).

## 3. Results

### 3.1. Patients Characteristics and In-Hospital Treatments

Of 93,377,054 hospitalizations from 2003 to 2014, 13,805 were admitted with salicylate intoxication as the primary diagnosis. The average age was 34 ± 19 years. Of these, 35.0% were male and 65.4% used salicylates for a suicidal attempt. During hospitalization, 5.5% and 5.9% required invasive mechanical ventilation and renal replacement therapy, respectively (Table 1).

### 3.2. Inpatient Prevalence of Salicylate Intoxication

Table 2 shows the annual inpatient prevalence of salicylate intoxication in hospitalized patients. The overall inpatient prevalence of salicylate intoxication was 147.8 cases per 1,000,000 admissions, ranging from 128.4 to 167.4 cases per 1,000,000 admissions during the study period.

### 3.3. Complications, Organ Failure, and In-Hospital Mortality

The most common complications of salicylate intoxication were electrolyte and acid-base disorders, including hypokalemia (25.4%), acidosis (19.1%), and alkalosis (11.1%). In total, 20.0% of hospitalized patients developed ≥1 organ failure. Kidney failure (9.3%) was the most common observed organ dysfunction, followed by respiratory (6.8%), neurological (5.0%), circulatory (3.5%), hematological (2.2%), and liver failure (0.8%) (Table 1).

Of 13,805 patients hospitalized for salicylate intoxication, 132 (1.0%) died in the hospital. Multivariable analysis identified that age ≥30, Asian/Pacific Islander race, diabetes mellitus, hyponatremia, ventricular arrhythmia, kidney failure, respiratory failure, and neurological failure were associated with higher in-hospital mortality, whereas African American and Hispanic race were associated with lower in-hospital mortality (Table 3).

Hospitalized patients who developed 0, 1, 2, and ≥3 organ failures had increased associated in-hospital mortality of 0.1%, 2.3%, 7.5%, 14.3%, respectively. The corresponding adjusted odds ratio for in-hospitality mortality associated with increasing number of organ failure consisted of 12.77 (95% CI 6.75–24.17) for one organ failure, 33.96 (95% CI 17.01–67.80) for two organ failures, and 54.16 (95% CI 25.07–116.98) for ≥3 organ failures.

### 3.4. Length of Hospital Stay and Hospitalization Cost

The median length of hospital stay was two (interquartile range 1–3) days. The median hospitalization cost per patient was USD 11,172 (interquartile range 6671–19,726) (Table 1).

## 4. Discussion

This is a large hospitalized cohort study investigating inpatient prevalence, characteristics, in-hospital outcomes, and resource use for patients admitted for salicylate intoxication. The overall inpatient prevalence of salicylate intoxication was 147.8 cases per 1,000,000 admissions. Among hospitalized salicylate intoxication patients, 35% of patients were male, and 65.4% used salicylate-containing drugs for suicidal attempts. The most common complication was hypokalemia, followed by acidosis and alkalosis, respectively. The in-hospital mortality rate was 1%. We identified independent factors associated with increased in-hospital mortality, including ventricular arrhythmia/cardiac arrest, respiratory failure, age ≥30, Asian/Pacific Islander race, hyponatremia, diabetes mellitus, kidney failure, and neurological failure. Kidney failure was the most common observed organ dysfunction. Not unexpectedly, the number of organ failures was progressively related with higher in-hospital mortality.

In 2018, the American Association of Poison Control Centers reported that approximately 27,000 patients were exposed to salicylates in either acetylsalicylate or methyl salicylate formulations [2]. However, this report included all age ranges and patients in the outpatient setting. Therefore, it is not unsurprising that their number of reported patients with salicylate intoxication is higher than in our study composed of a cohort of adult hospitalized patients. Since most salicylate-containing products are over-the-counter medications, salicylate intoxication remains one of the most common drug overdoses in the United States. The rate of reported intentional overdose of salicylates in the United States was 52.8% of salicylate intoxication patients, comparable to our study [2].

Salicylate intoxication can result in several complications, particularly of electrolyte and metabolic abnormalities. Our study demonstrated that the most common complication found in salicylate intoxication was hypokalemia, followed by acidosis and alkalosis. Hypokalemia in salicylate intoxication may be primarily contributed to by an alkalemia driven by respiratory alkalosis, the hallmark acid-base abnormality found in salicylate intoxication. The alkaline pH shifts potassium into the intracellular fluid and enhances distal tubule renal excretion of potassium [4,5,6,7,8]. Hypokalemia in turn can further complicate salicylate intoxication and its management. For instance, it can contribute to cardiac arrhythmias or deter urine alkalization [1,9]. Salicylate intoxication can result in either alkalemia or acidemia due its associated opposing acid-base pathophysiologic factors, often resulting in mixed underlying acid-base abnormalities. Respiratory alkalosis with alkalemia can be initially observed on presentation due to direct stimulation of the central medulla respiratory center by salicylates [1]. A previous study reported that respiratory alkalosis was associated in 78% of salicylate ingestions alone and 40% of a combination of salicylate and other drug intoxications [10]. Acidemia may occur as salicylates can interfere with carbohydrate metabolism and uncoupling of mitochondrial oxidative phosphorylation, thus causing a high anion gap metabolic acidosis from ketone and lactic acid accumulation [9]. A minor component of the metabolic acidosis can also contributed to by salicylate itself, and the increase in renal bicarbonate excretion in response to the respiratory alkalosis [3]. The severity of acidemia can be further exacerbated when the compensatory respiratory hyperventilation for a primary metabolic acidosis is suppressed by co-ingestion of central nervous system depressant agents [1]. In pediatric patients, acidosis may more frequently dominate the acid-base pathophysiology. The combination of both a respiratory alkalosis and metabolic acidosis with salicylate intoxication can result in the presentation of a normal range serum pH, thus masking the underlying acid-base abnormalities. Iatrogenic bicarbonate therapy utilized to enhance salicylate elimination by urine alkalinization can also contribute to mixed acid-base balance disturbances.

The in-hospital mortality rate of salicylate intoxication was one percent. In 2018, the American Association of Poison Control Centers reported a salicylate intoxication mortality rate of 0.4% [2]. This discrepancy is, again, likely due to the inclusion of patients from the outpatient setting in their report, which tend to have less severe salicylate intoxications. Other previous studies have reported mortality rates of salicylate intoxication ranging from 4 to 10% [11,12,13,14]. Our study further identified independent factors associated with increased in-hospital mortality in hospitalized patients with salicylate intoxication. These risk factors included ventricular arrhythmias/cardiac arrest, respiratory failure, age ≥30, Asian/Pacific Islander race, hyponatremia, diabetes mellitus, kidney failure, and neurological failure. Ventricular arrhythmias and cardiac arrest are critical, life-threatening conditions with high mortality outcomes. Cardiac arrhythmias are not common in salicylate intoxications, particularly ventricular arrhythmias. In a previous study, ventricular arrhythmias were identified in fatal cases only [14]. Higher mortality rates were also demonstrated in a cohort of intubated salicylate intoxication patients, suggesting that respiratory failure is associated with increased mortality [15]. Indeed, clinicians are warned that severe acidemia can occur after intubation for mechanical ventilation, particularly with sedation and neuromuscular relaxant. Suppression of the respiratory drive can result in uncompensated metabolic acidosis that leads to a rapid decrease in serum pH and increased toxicity as salicylates shift from serum into cells [16]. As such, it has been recommended that bicarbonate therapy can be given prior to intubation in an effort to maintain the serum pH [3]. Finally, other studies have similarly identified risk factors for increased mortality, including age, neurological symptoms, serum salicylate levels, delayed diagnosis, respiratory rate, and acidosis [11,12,14].

Organ failure is a well-known risk factor associated with mortality, particularly in critical illness [17]. Our study demonstrated that organ failure was observed in 20% of patients and identified kidney failure as the most common organ failure in salicylate intoxication. Kidney failure can be contributed to by multifactorial etiologies in salicylate intoxication. Salicylates can directly cause renal damage by inducing acute tubular necrosis, acute interstitial nephritis, papillary necrosis, and nephrotic syndrome [1]. In addition, salicylate intoxication frequently results in volume depletion via increased insensible loss (e.g., fevers, tachypnea), poor oral intake, and vomiting, resulting in hypovolemic kidney failure [1]. Our study demonstrated that kidney failure and neurological failure was associated with increased in-hospital mortality. Our study also supported that an increase in the number of organ failures was progressively associated with higher in-hospital mortality. This finding confirmed the impact of multi-organ failures on outcome [18].

This study was composed of a large hospitalized cohort of salicylate intoxication patients, but there were some limitations. Since NIS is a hospitalized-base cohort, our study did not include patients who were presented to the outpatient, urgent care, or emergency departments and were treated but not admitted. Hence, the prevalence and mortality rate of salicylate intoxication only reflects those with more severe disease requiring hospitalization. Secondly, due to the NIS structure, we did not evaluate long-term outcomes. Thirdly, we did not address some potential factors that might affect outcomes, such as serum salicylate levels, route of salicylate administration, and data on the co-ingestion of other drugs.

In conclusion, hospitalization for salicylate intoxication occurred in 148 per 1,000,000 admissions in the United States. Kidney failure was the most common observed organ dysfunction. One percent of hospitalized patients for salicylate intoxication died in hospital. Several factors were identified to be associated with in-hospital mortality.

## Figures and Tables

**Table 1 jcm-09-02638-t001:** Clinical characteristics, in-hospital treatments, complications, outcomes, and resource use for hospitalized patients with salicylate intoxication.

	Total
Clinical Characteristics	
N (%)	13,805
Age (years), mean ± SD	34.0 ± 18.7
<20	3902 (28.3)
20–29	3228 (23.4)
30–39	1951 (14.1)
≥40	4710 (34.2)
Male	4811 (35.0)
Race	
Caucasian	7729 (56.0)
African American	1391 (10.1)
Hispanic	1311 (9.5)
Asian or Pacific Islander	200 (1.4)
Other	3174 (23.0)
Year of Hospitalization	
2003–2006	5011 (36.3)
2007–2010	4434 (32.1)
2011–2014	4360 (31.6)
Alcohol drinking	2216 (16.1)
Use salicylate for suicide	9032 (65.4)
Concurrent analgesics overdose	967 (7.0)
Concurrent psychotropic medication overdose	896 (6.5)
Obesity	521 (3.8)
Diabetes Mellitus	801 (5.8)
Hypertension	2137 (15.5)
Dyslipidemia	749 (5.4)
Coronary artery disease	512 (3.7)
Congestive heart failure	239 (1.7)
Atrial flutter/fibrillation	172 (1.2)
Chronic kidney disease	218 (1.6)
Liver cirrhosis	116 (0.8)
Treatments	
Gastric lavage	344 (2.5)
Non–invasive ventilation	64 (0.5)
Invasive mechanical ventilation	760 (5.5)
Blood component transfusion	356 (2.6)
Renal replacement therapy	811 (5.9)
Complications and Outcomes	
Hypoglycemia	127 (0.9)
Hyponatremia	262 (1.9)
Hypernatremia	281 (2.0)
Hypokalemia	3508 (25.4)
Hyperkalemia	148 (1.1)
Acidosis	2641 (19.1)
Alkalosis	1527 (11.1)
Rhabdomyolysis	258 (1.9)
Seizure	565 (4.1)
Gastrointestinal bleeding	363 (2.6)
Sepsis	126 (0.9)
Ventricular arrhythmia/cardiac arrest	95 (0.7)
Kidney failure	1279 (9.3)
Respiratory failure	943 (6.8)
Circulatory failure	484 (3.5)
Liver failure	110 (0.8)
Neurological failure	689 (5.0)
Hematological failure	303 (2.2)
In–hospital mortality	132 (1.0)
Resource Use	
Length of hospital stay (days), median, interquartile range	2 (1–3)
Hospitalization cost (USD), median, interquartile range	11,172 (6671–19,726)

**Table 2 jcm-09-02638-t002:** Distribution and inpatient prevalence of salicylate intoxication from 2003 to 2014.

Year	Total Number of Admissions for Salicylate Intoxication	Total Number of Admissions	Inpatient Prevalence (per 1,000,000 Admissions)
2003	1311	7,977,728	164.3
2004	1340	8,004,571	167.4
2005	1240	7,995,048	155.1
2006	1120	8,074,825	138.7
2007	1177	8,043,415	146.3
2008	1153	8,158,381	141.3
2009	1003	7,810,762	128.4
2010	1101	7,800,441	141.1
2011	1145	8,023,590	142.7
2012	1097	7,296,968	150.3
2013	1053	7,119,563	147.9
2014	1065	7,071,762	150.6
Total	13,805	93,377,054	147.8

**Table 3 jcm-09-02638-t003:** Factors associated with in-hospital mortality for salicylate intoxication.

Variables	Univariable Analysis	Multivariable Analysis
Crude Odds Ratio (95% CI)	*p-*Value	Adjusted Odds Ratio (95% CI)	*p-*Value
Age (years)				
<20	1 (reference)		1 (reference)	
20–29	6.89 (2.02–23.52)	0.002	3.40 (0.92–12.60)	0.07
30–39	14.85 (4.44–49.66)	<0.001	5.14 (1.43–18.46)	0.01
≥40	25.33 (8.01–80.08)	<0.001	3.98 (1.17–13.54)	0.03
Male	1.77 (1.25–2.49)	0.001		
Race				
Caucasian	1 (reference)		1 (reference)	
African American	0.25 (0.09–0.68)	0.007	0.29 (0.09–0.92)	0.04
Hispanic	0.13 (0.03–0.54)	0.005	0.11 (0.02–0.61)	0.01
Asian or Pacific Islander	1.78 (0.65–4.89)	0.27	3.45 (1.16–10.29)	0.03
Other	0.94 (0.63–1.40)	0.77	0.85 (0.52–1.41)	0.53
Year of data collection				
2003–2006	1 (reference)			
2007–2010	1.49 (0.99–2.24)	0.06		
2011–2014	1.04 (0.66–1.62)	0.88		
Alcoholic drinking	1.10 (0.70–1.73)	0.67		
Use salicylate for suicide	0.63 (0.45–0.89)	0.009		
concurrent analgesics overdose	0.52 (0.21–1.27)	0.15		
concurrent psychotropic agent overdose	1.31 (0.71–2.44)	0.39		
Obesity	1.00 (0.41–2.46)	0.99		
Diabetes Mellitus	2.43 (1.45–4.06)	0.001	2.26 (1.26–4.09)	0.007
Hypertension	1.48 (0.97–2.25)	0.07		
Dyslipidemia	2.09 (1.20–3.66)	0.01		
Coronary artery disease	5.10 (3.17–8.20)	<0.001		
Congestive heart failure	7.11 (4.02–12.56)	<0.001		
Atrial fibrillation/flutter	3.18 (1.28–7.87)	0.01		
Chronic kidney disease	0.47 (0.07–3.39)	0.46		
Liver cirrhosis	4.80 (1.93–11.96)	0.001		
Hypoglycemia	0.82 (0.11–5.90)	0.84		
Hyponatremia	4.87 (2.60–9.14)	<0.001	3.16 (1.52–6.57)	0.002
Hypernatremia	4.52 (2.41–8.48)	<0.001		
Hypokalemia	0.68 (0.44–1.06)	0.09		
Hyperkalemia	6.22 (2.99–12.97)	<0.001		
Acidosis	3.26 (2.30–4.61)	<0.001		
Alkalosis	1.03 (0.60–1.77)	0.91		
Rhabdomyolysis	5.98 (3.32–10.74)	<0.001		
Seizure	3.80 (2.29–6.29)	<0.001		
Gastrointestinal bleeding	2.09 (0.97–4.51)	0.06		
Sepsis	16.97 (9.74–29.57)	<0.001		
Ventricular arrhythmia/cardiac arrest	133.28 (84.60–209.96)	<0.001	54.42 (29.46–100.52)	<0.001
Kidney failure	7.25 (5.10–10.30)	<0.001	1.84 (1.18–2.87)	0.007
Respiratory failure	59.43 (38.71–91.24)	<0.001	26.61 (16.41–43.14)	<0.001
Circulatory failure	7.08 (4.56–10.98)	<0.001		
Liver failure	12.44 (6.51–23.78)	<0.001		
Neurological failure	6.09 (4.04–9.17)	<0.001	1.73 (1.04–2.88)	0.04
Hematological failure	5.48 (3.11–9.66)	<0.001

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
