# Peer review of "Hospitalizations for Acute Salicylate Intoxication in the United States"

_jcm, 2020, doi:10.3390/jcm9082638_

Round 1

Reviewer 1 Report

The study is interesting since it deals with one of the most common intoxications. The provided data are interesting, and the manuscript is well-written. Also, the discussion offers in a compact way many interesting pathophysiological and therapeutic information. Importantly, besides providing new information, the document is educative.

Author Response

Response to Reviewer #1

The study is interesting since it deals with one of the most common intoxications. The provided data are interesting, and the manuscript is well-written. Also, the discussion offers in a compact way many interesting pathophysiological and therapeutic information. Importantly, besides providing new information, the document is educative.

Response: We thank you for reviewing our manuscript and for your critical evaluation.

All authors thank the Editors and reviewers for their valuable suggestions. The manuscript has been improved considerably by the suggested revisions

Reviewer 2 Report

This is an interesting evaluation of hospitalization and outcome of salicylate intoxication. The objective is clearly stated, the approach scientifically sound, the results clearly presented and the conclusions to the point.  The authors also clearly listed the limitations of their study. This information is succinctly presented. I have the following suggestions/questions/comments:

  1. Is there any high-level information on the products at the origin of the intoxication?  Example: oral vs. topical (I am assuming almost all oral). Any impact on the endpoints evaluated?
  2. The introduction appropriately discusses acute versus chronic intoxication, but it would be fair to indicate that in the current study most cases will reflect an acute intoxication state. To that point, would it be better to add "acute" in the title and objective?
  3. A paragraph in the introduction summarizing the metabolism pathway of salicylates in humans would be useful.
  4. Likewise, I suggest adding 1-2 sentences describing the mechanism of toxicity (i.e. effect on oxidative phosphorylation).

Author Response

Response to Reviewer #2

This is an interesting evaluation of hospitalization and outcome of salicylate intoxication. The objective is clearly stated, the approach scientifically sound, the results clearly presented and the conclusions to the point.  The authors also clearly listed the limitations of their study. This information is succinctly presented. I have the following suggestions/questions/comments:

Response: We thank you for reviewing our manuscript and for your critical evaluation.

Comment #1

Is there any high-level information on the products at the origin of the intoxication?  Example: oral vs. topical (I am assuming almost all oral). Any impact on the endpoints evaluated?

Response: The reviewer raises very important point. The national inpatient sample database did not have data on route of salicylate administration. The following statements have been added to limitation section.

“We did not address some potential factors that might affect outcomes, such as serum salicylate levels, route of salicylate administration, and data on co-ingestion of other drugs.”

Comment #2

The introduction appropriately discusses acute versus chronic intoxication, but it would be fair to indicate that in the current study most cases will reflect an acute intoxication state. To that point, would it be better to add "acute" in the title and objective?

Response: We agree with the reviewer. These has been revised as suggested

Comment #3

A paragraph in the introduction summarizing the metabolism pathway of salicylates in humans would be useful.

Response: We agree with the reviewer. The following statements have been added to describe the metabolism pathway of salicylates in humans.

“After ingestion, salicylates are rapidly absorbed in the stomach. Salicylates are mostly protein bound and therefore limited to intravascular space. Salicylates are metabolized via several different pathways in the liver, including glycination into salicyluric acid, which is both less toxic and more rapidly excreted by kidney. Only a small amount of salicylates is excreted unchanged in the urine.”

Comment #3

Likewise, I suggest adding 1-2 sentences describing the mechanism of toxicity (i.e. effect on oxidative phosphorylation).

Response: We appreciate the reviewer's input. The following statements have been added to describe the mechanism of salicylate intoxication.

“Salicylates directly or indirectly affect most organ systems by uncoupling oxidative phosphorylation, inhibiting Krebs cycle enzymes, and inhibiting amino acid synthesis.”

All authors thank the Editors and reviewers for their valuable suggestions. The manuscript has been improved considerably by the suggested revisions! 

Reviewer 3 Report

This manuscript is describing inpatient prevalence, characteristics, in-hospital outcomes, and resource use for salicylate intoxication hospitalizations in the United during the period of 2003 to 2014. This is a large and comprehensive cohort study of salicylate intoxication patients that provides useful overview and analyses on parameters associated with in-hospital mortality, and manuscript is well-written. Although this is limited to in-hospital patients of salicylate intoxication, authors efficiently described limitations of the present study in the Discussion.

I have a question that authors need to address or clarify as follows.

  1. It is recommended to include the statement explaining the basis of selecting the period of 2003 to 2014 for the analyses. I am wondering why authors did not include more recent years.

Author Response

Response to Reviewer #3

This manuscript is describing inpatient prevalence, characteristics, in-hospital outcomes, and resource use for salicylate intoxication hospitalizations in the United during the period of 2003 to 2014. This is a large and comprehensive cohort study of salicylate intoxication patients that provides useful overview and analyses on parameters associated with in-hospital mortality, and manuscript is well-written. Although this is limited to in-hospital patients of salicylate intoxication, authors efficiently described limitations of the present study in the Discussion.

Response: We thank you for reviewing our manuscript and for your critical evaluation.

Comment #1

It is recommended to include the statement explaining the basis of selecting the period of 2003 to 2014 for the analyses. I am wondering why authors did not include more recent years.

Response: We appreciate the reviewer’s important input. This study utilized the national inpatient sample (NIS) dataset for the analysis. We purchased this dataset from the Healthcare Cost and Utilization Project (HCUP) of the Agency for Healthcare Research and Quality (AHRQ) in 2019. At the time of purchase, the NIS data was available only up to year 2014.

All authors thank the Editors and reviewers for their valuable suggestions. The manuscript has been improved considerably by the suggested revisions! 
